# Gaitor: Learning a Unified Representation for Continuous Gait Transition and Terrain Traversal for Quadruped Robots

## Abstract

The current state-of-the-art in quadruped locomotion is able to produce robust motion for terrain traversal but requires the segmentation of a desired trajectory into a discrete set of skills such as trot, crawl and pace. This misses the opportunity to leverage commonalities between individual gait types for efficient learning and are unable to smoothly transition between them. Here, we present *Gaitor*, which creates a learnt representation capturing correlations across multiple distinct gait types resulting in the discovery of smooth transitions between motions. In particular, this representation is compact meaning that information common to all gait types is shared. The emerging structure is interpretable in that it encodes phase correlations between the different gait types which can be leveraged to produce smooth gait transitions. In addition, foot swing characteristics are disentangled and directly addressable. Together with a rudimentary terrain encoding and a learned planner operating in this structured latent representation, *Gaitor* is able to take motion commands including gait type and characteristics from a user while reacting to uneven terrain. We evaluate *Gaitor* in both simulated and real-world settings, such as climbing over raised platforms, on the ANYmal C platform. To the best of our knowledge, this is the first work learning an interpretable unified-latent representation for *multiple* gaits, resulting in smooth and natural looking gait transitions between trot and crawl on a real quadruped robot.

## 1 Introduction

With the advances in optimal control (Mastalli et al., 2020; 2022a) and reinforcement learning (RL) (Hoeller et al., 2023; Rudin et al., 2022a) methodologies in recent years, quadruped robots are realising robust and efficient locomotion over uneven terrain. This has made quadrupeds an excellent choice for the execution of inspection and monitoring tasks, search and rescue, and delivery tasks. Despite the effectiveness of modern controllers, the resulting methodologies utilise a distinct set of discrete skills (gaits) to generate locomotion trajectories. These skills are independently learnt and smooth transitions between these motions is impossible. Thus, commonalities between specific gaits are ignored leading to inefficient representations. In some cases, core traits common to multiple skills are relearned. For example, the trot and pace gaits are both defined by the coordinated movement of pairs of legs in phase with each other. This correlation could be leveraged to represent both gaits in an efficient manner.

Inspired by recent advances in planning in structured latent-spaces for quadruped locomotion, we posit that continuous transition between gaits is achievable by learning a single latent representation for multiple robot skills. This is realised by utilising a generative model exposed to discrete examples of each gait type, specifically trot, crawl, pace and bound. A variational auto-encoder (VAE) (Kingma & Welling, 2014; Rezende et al., 2014) captures correlations between each gait creating a two-dimensional space in which information from each skill is shared across the representation of each gait. For example, trot, crawl and pace are captured in a two-dimensional plane, and a smooth transition from trot to crawl to pace and back again is achievable. The VAE is trained purely with expert demonstrations from simulation in a self-supervised manner using the $\beta$-VAE loss. Crucially, this loss is the evidence lower-bound (ELBO), which means that there is no interaction with the environment during training.

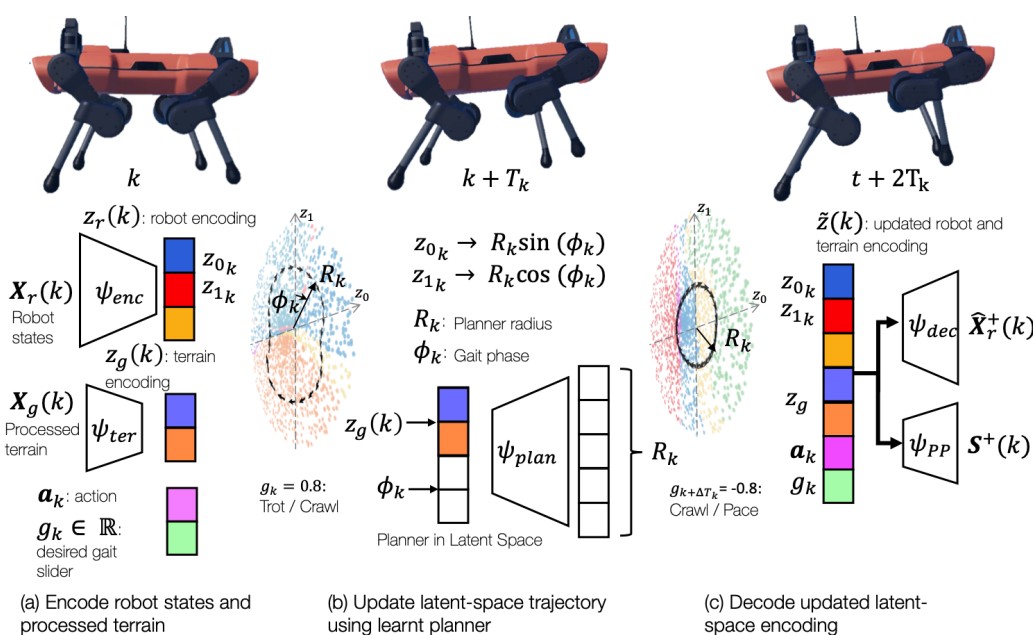

Figure 1: This approach, *Gaitor*, solves for a variety of locomotion tasks by learning a unified representation over robot motions. We learn a single representation $\mathbf{z}_r$ for multiple gaits (trot, pace and crawl), which is conditioned on terrain features $\mathbf{z}_g$. These are inferred using the encoders in panel (a). The latent-representation $\mathbf{z}_r$ forms a compact solution-space to a continuous set of different locomotion-problems meaning that new intermediate gaits are automatically discovered by traversing the space. To select a particular locomotion trajectory, we traverse the trajectory using oscillatory signals injected into two latent-space dimensions $z_0$ and $z_1$, see panel (b). A learnt model parameterises the radius $R_k$ of the oscillatory signals to adjust the robot's swing characteristics to react to uneven terrain. The latent-space paths are decoded to create joint-space trajectories to control the robot as shown in panel (c).

The VAE is able to infer this transition leading to novel intermediate gaits not seen during training. This is achieved since the latent space captures the coordination between front and rear pairs of legs and infers the phase between them. For example, consider just the front legs for any of the gaits apart from bound. For these gaits, the right and left legs are out of phase: if the left leg is in swing the right is in support. The rear pair of legs are similarly out of phase with each other for all the gaits apart from bound. To generate a specific gait such as pace, the front right leg must be in phase with the rear right leg. For trot the legs on the same side of the robot are out of phase, and for crawl, they are half way out of phase. Crucially, the VAE is able to infer the phase between the front and rear leg pairs during training and captures the correlations between these gaits. The result is a compact representation in which seamless transitions between these three gaits can be found. The structure of the latent space physically transforms as the gait transitions meaning that the same trajectory in latent space decodes to any of the following gaits and their intermediaries: trot, crawl or pace.

The latent-space representation not only encodes multiple gaits, but also is conditioned on perceptive information for terrain traversal. The evolution of the terrain in front of the robot is encoded and conditions the latent state to inform the future trajectories. However, the nominal swing heights and lengths may not be sufficient for traversing uneven terrain. For example, as one is walking it is necessary to increase the footswing height or length to clear a particular obstacle such as a slightly high pavement or step. This is an intuitive behaviour which we experience often. Similarly, the latent-space representation for the entire gait is also intuitively understandable and as discovered in *VAE-Loco* (Mitchell et al., 2023) locomotion trajectories encoded into the space are oscillatory. Indeed, a large vertical amplitude leads to longer footswings and a displacement in the horizontal decodes to a higher swing apex. This structure is leveraged via a learnt planner in latent space, which continuously adjusts the amplitudes of the two oscillatory signals as the robot walks and encounters terrain. These adjustments react to terrain resulting in larger and longer steps to clear an obstacle and shorter more considered footfalls whilst the robot is climbing terrain.

In this paper, we present *Gaitor* a unified representation for quadruped locomotion. This space efficiently captures the trot and crawl gaits as well as facilitating smooth transitions between each despite being trained without transitions. Furthermore, the resulting locomotion is conditioned on the terrain ahead of the robot via an encoder, whilst a planner in latent space makes fine adjustments to the robot's swing heights and lengths. The VAE and the planner are packaged together to form a trajectory optimiser in an MPC-style control loop. This is used to demonstrate terrain traversal as well as continuous gait transitions and are deployed on the real ANYmal C robot (Hutter et al., 2016). The resulting locomotion and the mutation of the latent-space trajectory and structure are analysed. To the best of our knowledge, this is the first work showing smooth and continuous transitions between distinct gaits.

## 2 RELATED WORK

Recent advances in quadruped locomotion literature is driving the utility of these platforms. Reinforcement learning (RL) methods (Gangapurwala et al., 2020; 2022; Rudin et al., 2022b) in particular are growing increasingly popular methods for locomotion due to advances in simulation capabilities. A recent trend for RL is to solve locomoton by learning a set of discrete skills with a high-level governor. An example of this is in Hoeller et al. (2023), which learns a set of distinct skills designed to tackle a set of sub-tasks such as traverse a gap before learning a selector which chooses a distinct option for the task at hand. This produces impressive behaviour, but no information is shared across tasks meaning that each skill learns information from scratch without leveraging prior knowledge. Another impressive work *Locomotion-Transformer* (Caluwaerts et al., 2023) learns a generalist policy for multiple locomotion skills. This approach uses a transformer model to learn a multi-modal policy conditioned on terrain. In contrast, *Gaitor* learns an explicit and compact 2D representation for the dynamics of multiple locomotion gaits. The representation captures the phase relationships between each leg and embeds these into the latent-space structure. As a result and uniquely to *Gaitor*, unseen intermediate gaits are automatically discovered by traversing this space.

Work by Yang et al. (2021) tackles gait switching by learning to predict the phases between each leg. The gait phase and the relative phases of each leg form this RL method's action space. The predicted phases create a contact schedule for a model predictive controller (MPC), which solves for the locomotion trajectory. In Yang et al. (2021), the gait phase is an inductive bias designed to explicitly determine the swing and stance properties of the gait. The utilisation of the gait phase as an inductive bias for locomotion contact scheduling is common in literature. For example, work by Wu et al. (2023) conducted contemporaneously to *Gaitor* utilises a latent-space in which different gaits/skills are embedded. A gait generator operates from this space and selects a particular gait to solve part of the locomotion problem. This utilises a series of low-level skills prior to learning the latent representation and skills are trained using pre-determined contact schedules. In contrast, *Gaitor* learns to infer the relationships between gait types and automatically discovers the intermediate skills required for continuous gait transition purely from examples of the discrete gaits.

The utilisation of a learnt latent-representations for locomotion is a growing field of interest. These methods are particularly efficient at learning compact representations, which form solution spaces for locomotion generation. For example, (Li et al., 2020) learns high-level latent-action space and a low-level policy from expert demonstrations using imitation learning. Locomotion results via an MPC formulation, which is solved via random shooting using a learnt dynamics model. In a similar vein, *VAE-Loco* (Mitchell et al., 2023) creates structured latent-space where the characteristics of the trot gait are disentangled in the space. This structure is exploited leading to locomotion where the cadence, footstep height and length can be varied continuously during the swing phase. *VAE-Loco* is constrained to a single gait and operation on flat ground only. Nevertheless, this work forms the inspiration for *Gaitor* which tackles multiple gaits and perceptive locomotion on uneven terrain.

## 3 APPROACH

This approach, *Gaitor*, solves for a variety of locomotion tasks including continuous gait transitions between trot and crawl via unseen intermediate gait types as well as perceptive terrain-traversal. To do this, we create a single learnt-representation for all the gaits we wish to use. The learnt

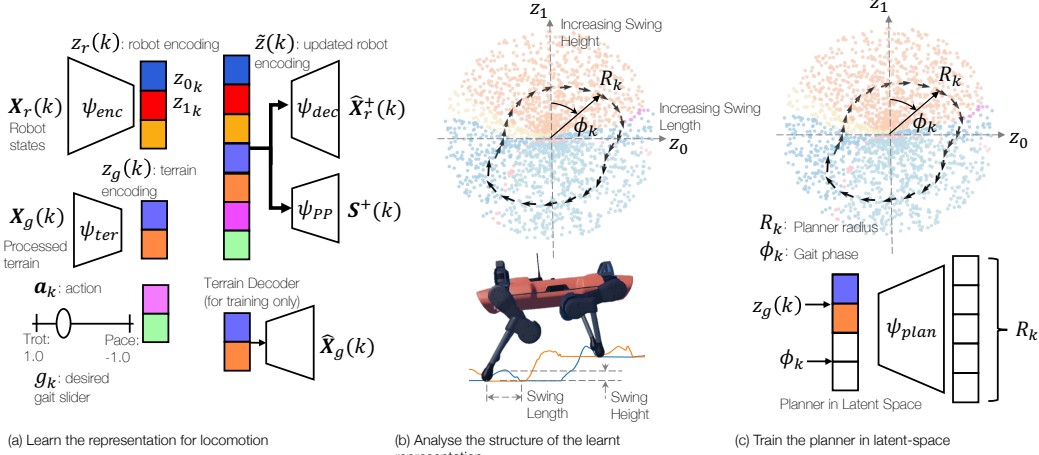

(a) Learn the representation for locomotion

(b) Analyse the structure of the learnt representation

(c) Train the planner in latent-space

Figure 2: The training process is divided into three stages. The first stage in panel (a) creates the learnt representations for the robot's dynamics $\mathbf{z}_r$ and for the terrain $\mathbf{z}_g$. Next, this representation is inspected to find useful structure. The latent dimensions $z_0$ and $z_1$ are plotted and in panel (b). We see that the magnitude of oscillations in $z_0$ and $z_1$ control footswing height and length respectfully. In panel (c), we learn a model to control these magnitudes using a planner in polar coordinates. The user-commanded gait phase $\phi_k$ is the planner's input and the radius $R_k$ is its output.

latent-representation can be thought of as a continuous set of solutions to the dynamics equation for robot locomotion. To control the robot, we must select a single solution. We traverse the latent representation to create a trajectory which we decode to generate the joint-space solution. In this paper, we use another learnt model to traverse the latent space and create locomotion trajectories.

There is a three stage training process for this approach followed by deployment. The first stage as shown in Fig. 2 (a) creates a representation for the dynamics for the multiple gaits and is conditioned using a terrain encoding, base-velocity action, and gait label. The second stage of the training process inspects the structure of the learnt representation. This structure is depicted in Fig. 2 (b) and informs the design of latent-space planner shown in Fig. 2 (c). The third stage is a behavioural cloning training regime to learn a planner designed to traverse the latent representation. Finally, once the latent space and its planner are trained, we deploy the approach in a model predictive control (MPC) framework. All components are trained using expert trajectories generated using *RLOC* (Gangapurwala et al., 2022) and exclude transitions between gaits.

## 3.1 Creating the Learnt Representation for Locomotion

The robot latent-space is created using a variational autoencoder (VAE) and an autoencoder (AE) for the robot states and the terrain encoding.

**Robot Encoding:** The inputs to the VAE are the robot states. The state at time step $k$ is constructed as $\mathbf{x}_k = [\mathbf{q}_k, \mathbf{ee}_k, \tau_k, \lambda_k, \dot{\mathbf{c}}_k, \Delta\mathbf{c}_k]$ and represent the joint angles, end-effector positions in the base frame, joint torques, contact forces, base velocity, gravity vector, roll and pitch of the base respectfully. The VAE's encoder $\psi_{enc}$ takes a history of these states as input. The frequency of these samples is $f_{enc}$ and is in general four to eight times lower than the decoder frequency $f_{dec}$. The input to the VAE is $\mathbf{X}_k$ and is constructed using $N$ past samples stacked together into a matrix. Similarly the output $\hat{\mathbf{X}}_k^+$ is made from $M$ predictions into the future from time step $k$. The robot's base is floating in space and must be tracked so that the decoded trajectories can be represented in the world frame. The encoder tracks the evolution of the base pose from a fixed coordinate: this coordinate is the earliest time point in the encoder input and all input poses are relative to this frame. All pose orientations are represented in tangent space. This is a common representation for the Lie-Algebra of 3D rotations since the non-linear composition of transformations is vector algebra in the this space, see Solà et al. (2021); Mastalli et al. (2022b) as examples. The output of the encoder parameterises a Gaussian distribution to create an estimate for the robot latent $\mathbf{z}_r(k)$.

The decoder predicts the future states $\hat{\mathbf{X}}_k^+$ at the control frequency $f_{dec} = f_{ctrl}$. This includes the base-pose evolutions, which are expressed relative to the base pose at the previous time-step. The inputs to the decoder are $\mathbf{z}_r(k)$ the desired action $\mathbf{a}_k$, gait $g_k$, and terrain encoding $\mathbf{z}_g(k)$. The gait slider $g_k \in \mathbb{R}$ is a continuous value used to select the gait or intermediate gait type. Distinct gaits such as trot, crawl, and pace are selected when $g_k = [1, 0, -1]$ respectfully.

The performance predictor (PP) predicts the contact states of the feet $\mathbf{S}_k$ and takes the robot encoding, and the desired action and gait as input. The PP has two functions: firstly during training the binary cross-entropy loss (BCE) is backpropagted through the VAE's encoder to structure the space (see *VAE-Loco* (Mitchell et al., 2023) for complete details). Secondly, the contact states are predicted during operation and sent to the tracking controller to enforce the contact dynamics.

**Terrain Encoding:** The terrain encoding is required so that the decoder and planner can react to terrain height variations. The heights $h(k)$ of the terrains are processed before being encoding and this pipeline is depicted in Fig. 3. The robot provides a 2.5 dimensional height map of the terrain around the robot constructed from depth images from four cameras around the robot. This raw map is filtered and in-painted in a similar fashion to Mattamala et al. (2022). The future footholds are estimated using forward kinematics and previous joint-states. We define the control pitch $\theta_c$ as the relative heights of terrain between the front feet and the rears across the body, see Fig. 3 (c). The control pitch is discontinuous and varies only when the robot takes a step. To convert this discontinuous variable to a continuous one, we filter it using a second-order linear time-invariant (LTI) filter. This converts the step-like control pitch sequence into a continuous time varying signal $\mathbf{X}_g(k)$ of the type shown in Fig. 3 (d). This is achieved by parameterising the resonant peak to occur at the peak of the foot swing and a damping factor of $0.5$. The input to the terrain encoder $\psi_{\mathrm{TER}}$ is a matrix of $N$ stacked values $\mathbf{X}_g(k)$ sampled at the control frequency $f_c$.

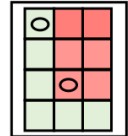

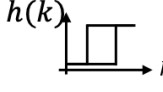

(a) Top-view of height map from vision

(b) Sample foot heights $h(k)$ for future steps

$$\boldsymbol{\theta}_c = \begin{bmatrix} h_{LF} - h_{RH} \\ h_{RF} - h_{LH} \end{bmatrix}$$

(c) Calculate control pitch $\theta_c$ from height samples

(d) Filter $\boldsymbol{\theta}_c$ using LTI filter

Figure 3: Terrain processing pipeline. The robot produces a 2.5D height map of the terrain around the robot. We process this map by sampling the heights at the footfall positions, see panels (a) and (b). The control-pitch angle $\theta_c$ is defined as the heights between the front and rear heights of the footfalls, see panel (c). These values are input to a second order resonant filter to create the input to the terrain encoder $\mathbf{x}_g$ (panel (d)).

**Training the VAE and Terrain AE:** The VAE and the terrain autoencoder are trained concurrently to create the learned representation. The VAE is optimised using gradients from the ELBO loss

$$\mathcal{L}_{\mathrm{ELBO}} = \mathrm{MSE}(\mathbf{X}_k^+, \hat{\mathbf{X}}_k^+) + \beta D_{\mathrm{KL}}[q(\mathbf{z}|\mathbf{X}_k)||p(\mathbf{z})], \tag{1}$$

and the binary cross-entropy loss creating

$$\mathcal{L}_{\mathrm{VAE}} = \mathcal{L}_{\mathrm{ELBO}} + \gamma \mathrm{BCE}(\mathbf{S}_k, \hat{\mathbf{S}}_k). \tag{2}$$

The VAE is optimised using GECO (Rezende & Viola, 2018) to tune the $\beta$ during training.

Simultaneously, the terrain encoder's weights are updated using the $\mathrm{MSE}_{\mathrm{VAE}}$ loss from the VAE and a mean-squared error between predicted $\hat{\mathbf{X}}_g(k)$ and the ground truth from the data $\mathbf{X}_g(k)$. This creates the terrain autoencoder loss

$$\mathcal{L}_{\mathrm{TER}} = \mathrm{MSE}(\mathbf{X}_r^+(k), \hat{\mathbf{X}}_r^+(k)) + \mathrm{MSE}(\mathbf{X}_g(k), \hat{\mathbf{X}}_g(k)). \tag{3}$$

The terrain decoder is only used for training purposes and is not required for deployment.

## 3.2 Inspecting the Latent Space

We analyse *Gaitor's* latent space to discover the useful structure within. To achieve this we use the analysis tools described in *VAE-Loco* Sec. V. A. In summary, this finds that the two latent dimensions with lowest variance $z_0$ and $z_1$ controls the robot's swing heights and lengths independently and that oscillatory trajectories injected into these dimensions decode to continuous locomotion. Points in these dimensions are plotted and colour coded by predicted stance $\mathbf{S}^+$ in Fig. 2 (b). The latent-space structure is explored fully in Sec. 4.

### 3.3 LATENT-SPACE PLANNER

The planner is designed to adjust the robot's swing length and height to help the robot traverse terrain. As shown in Fig. 2 (b), the planner design exploits the disentangled latent-space, and only controls $z_0$ and $z_1$ which affect the swing heights and lengths. The planner is parameterised in polar coordinates and take the gait phase $\phi(k)$ and terrain latent $\mathbf{z}_g(k)$ as input to predict the radius of the elliptical trajectory $R(k)$, see Fig. 2 (c). This trajectory updates the values of the two latents with lowest variance such that

$$z_0(k) \leftarrow R(k)sin(\phi(k)), \quad \text{and} \quad z_1(k) \leftarrow R(k)sin(\phi(k) + \pi/2). \tag{4}$$

**Training the Planner:** The planner in latent space is trained via behavioural cloning. Expert trajectories are encoded into the VAE producing latent trajectories, $\mathbf{z}_0^*, ..., \mathbf{z}_D^*$, where $D$ is the number of time steps in the encoded episode. Only the dimensions $z_0$ and $z_1$ are retained to create the dataset for the planner and are converted to polar coordinates. Thus, $z_0^*(0), ..., z_0^*(D)$ and $z_1^*(0), ..., z_1^*(D)$ are converted to radii $R^*(0), ..., R^*(D)$, and phase $\phi(0), ..., \phi(D)$. The latter and the terrain encoding $\mathbf{z}_k$ are inputs to the planner whilst the former is the desired output. To reproduce the large variations in the radius as the robot climbs steps, we utilise an architecture inspired by Ruiz et al. (2018) as seen in Fig. 2 (c). The planner predicts a probability distribution $r_c$ for the radius over $C$ discrete values via a softmax. This distribution is multiplied by the weight of each corresponding bin to estimate $R(k)$. The total loss is the sum between the MSE and the cross-entropy loss between the prediction $r_c$ and the label $r^*$:

$$\mathcal{L}_{\text{PLAN}} = \text{MSE}(R(k), R^*(k)) - \sum_{c=0}^{C-1} \log \left( \frac{\exp(r_c)}{\sum_{i=0}^{I-1} \exp(r_c)} r^* \right). \tag{5}$$

### 3.4 DEPLOYMENT OF *Gaitor*

The VAE, AE and its planner form a trajectory optimiser in a closed-loop control loop. Fig. 1 doubles as a deployment diagram. Starting with panel (a), we encode the history of robot states $\mathbf{X}_r(k)$ and the processed terrain input $\mathbf{X}_g(k)$ to estimate the latent states for the robot and terrain. Next, the planner overwrites the two most significant latent dimensions $z_0$ and $z_1$ as shown in panel (b). The operator has control over the robot's heading using the action $\mathbf{a}_k$, the desired gait $\mathbf{g}_k$ and the gait phase $\phi_k$. The concatenation of $\mathbf{z}_r$, $\mathbf{z}_g$, $\mathbf{a}_k$, and $\mathbf{g}_k$ are decoded to find the next joint states from $\hat{\mathbf{X}}_k^+$ and the future contact states $\mathbf{S}_k^+$. The next joint and contact states are sent to a whole-body controller (WBC) (Bellicoso et al., 2018). In practice, there is lag between the desired pitch and the robot's pitch. Therefore, we set the robot's base pitch to the average of the control pitch.

The entire approach operates on the CPU of the ANYmal C in a real-time control loop. As such there are strict time budget and the inference time is limited to the duration of one time tick, which for a control frequency of $400\,\text{Hz}$ is $2.5\,\text{ms}$. Therefore, we deploy Gaitor using a bespoke C++ implementation and vectorised code.

| Approach | RL vs Representation Learning | Discovers Intermediate Skills | Inferred Phase vs Phase as a Bias |
|---|---|---|---|
| (Caluwaerts et al., 2023) | RL | No | Inferred |
| (Hoeller et al., 2023) | RL | No | Inferred |
| (Yang et al., 2021) | RL + MPC | No | Inductive Bias |
| **Gaitor** | Rep. Learn | Yes | Inferred |

Table 1: The capabilities and formulations of a range locomotion policies are compared. All methods are capable of multiple gaits in order to traverse terrain. However, not all methods perceive their surrounds to achieve to tackle terrain. Inferred phase corresponds to whether the approach utilises the phase of the gait cycle in order to generate the motion types and is this inferred during training. In the first two methods, the gait-phase is either not implicitly inferred or an inductive bias. In (Yang et al., 2021), the phase of each and the offsets are an explicit actions rather than an inferred quantity.

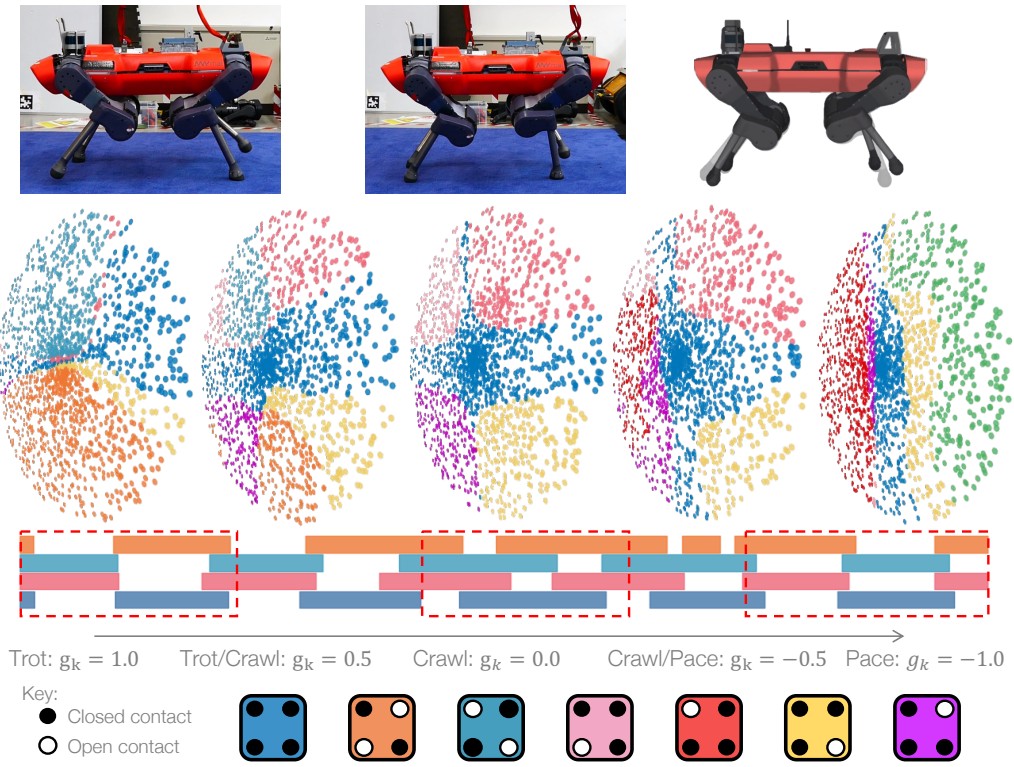

Figure 4: The contact state of each point in latent space changes as the gait slider moves from trot to crawl to pace. The contact schedule as the gait slider $g_k$ moves is depicted below the images of the latent spaces. The phase between the front and rear legs shifts from out-of-phase for trot to $90°$ for crawl and finally to in-phase for pace.

## 4 RESULTS AND ANALYSIS

Once *Gaitor* has been trained and deployed on the robot, its performance as a trajectory optimiser is evaluated and contextualised. Table 1 compares *Gaitor's* formulation to current art. To the best of our knowledge, there are few approaches which are capable of automatically discovering continuous transitions between locomotion gaits as the majority operate over discrete independent skills or use a suitably large model to learn a generalist policy conditioned on terrain as in Caluwaerts et al. (2023). The closest work Yang et al. (2021) utilises the gait phase as an inductive bias rather than inferring this from the data and is not perceptive, which limits its similarity to *Gaitor*. As such, we analyse *Gaitor* by posing the following guiding questions: 1) how do the latent spaces for each gait and the intermediate gaits look? 2) How does the *Gaitor* respond to uneven terrain? 3) How does the planner exploit the latent-space structure to help the robot traverse complex terrain? 4) Are the locomotion gaits and the intermediates feasible?

**Gait Switching:** The structure of the latent space is analysed as described in Sec. 3.2. As a result, we discover that trajectories in a 2D slice in latent space is sufficient to reconstruct high-fidelity locomotion trajectories. Each point in the space is colour coded to denote which feet are in contact with the terrain. Next, the gait labels are toggled such that each distinct gait (trot, crawl and pace) are selected. This produces three distinct 2D latent-spaces with sets of contact states.

We discover that it is possible to smoothly transition between all three gaits. The top row of Fig. 4 shows snapshots of the robot in each gait. For trot and crawl, these are images of the real robot as it transitions between gaits. However, we do not deploy the pace gait as the planner which produced the dataset trajectories are only stable when the robot is walking extremely slowly. Underneath the robot images are snapshots of the latent space. These images are synchronised with a contact schedule on the bottom row. Here, we see how the contact schedule shifts between each gait and the distinct sequences which make up each gait are highlighted. The latent-space trajectory is a symmetrical circle for all the gaits and it is only the latent-space morphology which changes.

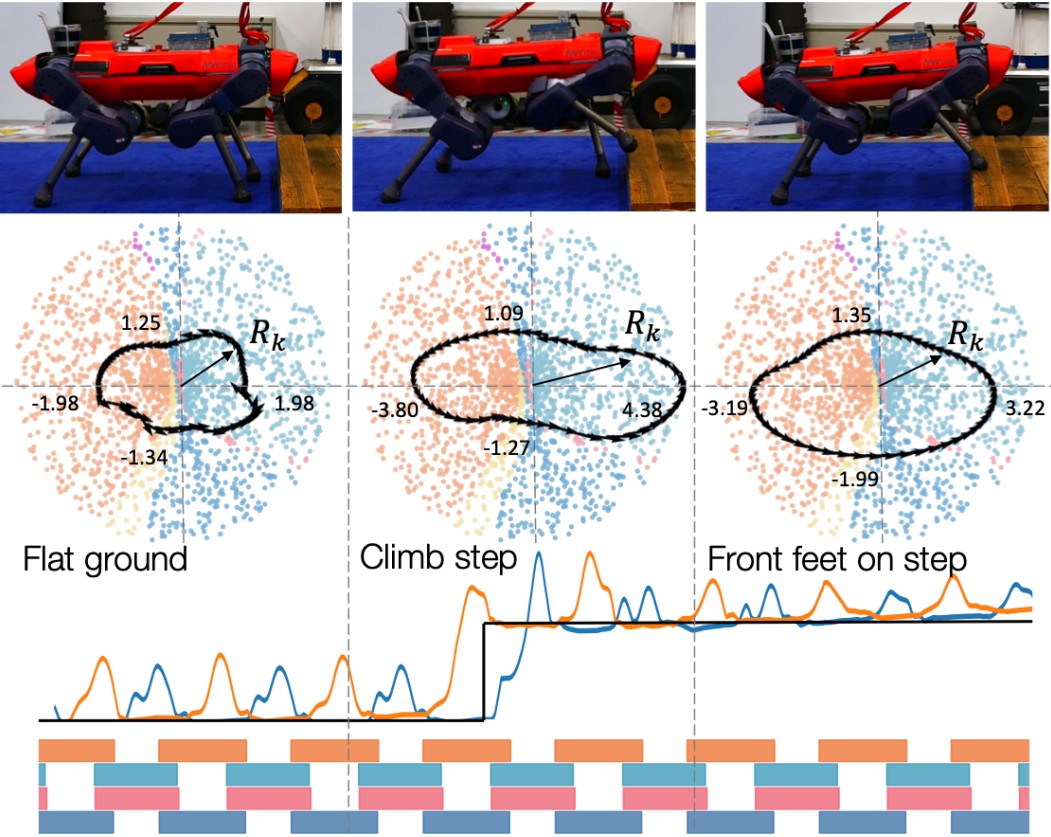

Figure 5: The robot walking onto a $12.5\,\mathrm{cm}$ platform. The VAE and planner in combination alters the locomotion trajectories in response to the terrain. The first phase is flat ground where the planner demands an expansive ellipse. When the robot steps onto the platform, the planner radius increases dramatically along the vertical axis, but is reduced along horizontal when compared to flat ground operation. This results in shorter footstep heights, but longer swing distances. Once both front feet are on the terrain, the latent-space trajectory returns to a circular path.

The gait transition must involve a single and specific sequence. For example, it is possible to transition from trot to crawl to pace and the reverse only. The resulting latent-space slices and the intermediary gait sequences are visible in Fig. 4. This is a result of the interpretable mechanism by which the gaits are represented. The VAE infers the relative phase between the front and rear pairs of legs. For example, if the left front and rear feet move phase, the gait is a pace. If they are out of phase, the gait is trot. As the gait label linearly increase from trot to crawl to pace, the relative phase between the front and rear legs shifts from out-of-phase (trot) to $90\,\mathrm{degrees}$ in-phase (crawl) and finally to in-phase (pace). Therefore, it is only possible to transition between gaits in this order and this explains how the intermediary gaits emerge. These emerge from the shifting in relative phase. This interpretation is similar to analytical analysis found in Tiseo et al. (2019), which views quadruped kinematics as a pair of bipeds linked by a rigid body. In that paper, gaits are generated by varying the phases between the front and rear bipeds similarly to what is inferred automatically by *Gaitor*.

**Terrain Traversal:** We demonstrate perceptive terrain-traversal using the highly dynamic trot gait successfully climb a $12.5\,\mathrm{cm}$ step. Fig. 5 shows the robot stepping up onto this step and is synchronised with the footstep heights of the front feet, the contact schedule and the planner output in latent space. The planner adjusts the locomotion in three distinct phases. The first is nominal locomotion on flat ground (phase I), the second is a short transient (phase II), where the front feet step up onto the palette and the final is a new steady state (phase III), where the robot has its front feet on the terrain and its rear feet on the ground.

In the flat-ground operation (phase I), the latent-space trajectory is broadly symmetrical. However, in phase II, the planner elongates the step-length placing the front feet over the lip of the step. The trajectory in phase II only lasts the duration required to swing the both front feet over terrain. For

|  | Trot | Trot (Climb Phase) | Crawl | Dynamic Crawl |
|---|---|---|---|---|
| RMSE [rad/s] | 0.031 | 0.069 | 0.101 | 0.131 |
| Swing Height [cm] | $8.3 \pm 0.58$ | $9.68 \pm 4.15$ | $5.42 \pm 0.65$ | $6.22 \pm 0.31$ |
| Swing Length [cm] | $10.4 \pm 0.53$ | $13.9 \pm 1.65$ | - | - |

Table 2: The feasibility and gait characteristics during four distinct modes of operation: trot on flat ground, trot whilst climbing the terrain, crawl gait, and the dynamic crawl gait. The root mean squared values (RMSE) are the average in joint space over a $5\,\mathrm{s}$ window of operation. All measurements are generated from experimental logs from real-robot experiments. The swing lengths are not measured for crawl and dynamic crawl as the robot is operated on the spot during these runs. Both trot data come from the terrain climb experiment.

the experiments in Fig. 5, the swing duration is set to $356\,\mathrm{ms}$. In phase III, the latent trajectory returns to a more symmetrical shape similar to the nominal flat ground trajectory seen in phase I, but is noticeably smoother.

**Exploiting Latent-Space Structure:** The planner in latent space modulates the robot's swing characteristics by exploiting the representation's structure. As mentioned, the structure is interpretable and disentangled meaning that footswing heights and lengths are independently modifiable as shown in Fig. 2 (b). We see this in action in Fig. 5, where the horizontal displacement of the planner's trajectory significantly increases as the robot initially climbs the step. On flat ground, the horizontal displacement in latent space is nominally 1.98 arbitrary units (au) and increases to 4.38 (au). We use arbitrary units in latent space, since the latent space has no units. This means that when *Gaitor* operates on flat ground, the swing length and heights are on average $(8.30 \pm 0.58)\,\mathrm{cm}$ and $(10.40 \pm 0.53)\,\mathrm{cm}$ respectully. During the climb, the planner extends the footswing height to $(9.68 \pm 4.15)\,\mathrm{cm}$ relative to the palette height and swing length to $(13.90 \pm 1.65)\,\mathrm{cm}$. Though there is a small increase in the mean swing height during the climb using the planner, this is statistically insignificant.

The planner in latent-space is designed to reproduce the encoded expert-trajectories in the latent space. As a result, it is the structure of the space which requires the additional planner to successfully traverse terrain. In essence, we see that circular latent-space trajectories with constant radii, as seen in right panel of Fig. 5, have constant swing heights and lengths irrespective of the terrain encoding. Thus in order to have a longer footswing as the robot climbs the step, the planner in latent-space is required to vary swing characteristics to traverse the terrain.

**Feasibility of the Trajectories:** All the terrain traversal, trot to crawl experiments are run successfully on the real ANYmal C robot. The true pace gait is not run on the real platform as the training data is only stable in a narrow window of operation. To measure the feasibility of the VAE's trajectories, we measure to what extent if any the WBC alters the trajectories. This is achieved by measuring the root mean-squared error (RMSE) between the *Gaitor's* trajectories and the commanded on the real robot in joint space. The RMSE is measured over a period of 2000 time steps equivalent to $5\,\mathrm{s}$ of operation during four distinct operation modes. The first mode is trot on flat ground; the second is during the climb up the terrain; the third is during a crawl; and the last is during a dynamic crawl where the rear foot makes contact as the front begins to swing. The results are summarised in the top row of Table 2. Here, we see that the WBC's commanded joint-space trajectories are extremely close to the VAE's predicted in all cases. We include a plot the WBC's commanded and VAE's output joint trajectories in the Appendix, see Sec. A.3. Hence, we conclude that the VAE's trajectories are dynamically feasible.

## 5  CONCLUSIONS

Inspired by the locomotion and the gaits seen in nature, we created a single latent-representation for gait transition and terrain traversal. This representation shares information across all the different gaits resulting in the generation of smooth and transitions between the skills via unseen intermediate gaits. In particular, the VAE infers the phase relationships between the front and rear pairs of legs for trot, crawl and pace. These gait representations are conditioned upon perceptive information to produce terrain-aware trajectories. However, this alone is insufficient for terrain traversal. A learnt planner in latent space adjusts the footstep heights and lengths in order to clear obstacles.

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

# A APPENDIX

## A.1 DATASET CREATION

As mentioned the dataset is created using *RLOC* (Gangapurwala et al., 2022). This learns a policy to estimate the swing characteristics to traverse uneven terrain which form constraints for the *Dynamic Gaits* (Bellicoso et al., 2018) trajectory optimiser. The inputs to the policy are the desired base heading $\mathbf{a}_k$ and the raw depth map provided by the robot.

We generate roughly $30\,\mathrm{min}$ of data of the robot traversing palettes of varying heights up to $12.5\,\mathrm{cm}$ in both trot and crawl. The pace gait is unable to traverse uneven terrain so is gathered on flat and is unable to pace stably at speeds greater than $0.1\,\mathrm{m/s}$.

## A.2 FILTER DESIGN FOR TERRAIN PROCESSING

The terrain inputs are processed as discussed in Sec. 3.1. As mentioned the terrain heights at locations of the future footfalls is queried from the terrain map and used to calculate the control pitch values. These are discontinuous and only update when the robot takes a step. To make these values continuous, we filter them using a second order resonant filter. As mentioned, we design the filter to have a resonant peak $t_p$ occurring when the in-swing foot is at its peak and to have a damping factor equal to $\zeta = 0.5$. As such, we express the continuous-time transfer function for the filter in the $s$-domain as:

$$\frac{Y(s)}{U(s)} = \frac{w_0}{s^2 + 2\zeta s w_0 + w_0^2}, \text{ where } w_0 = \frac{\pi}{t_p \sqrt{1 - \zeta^2}}. \tag{6}$$

The input to the transfer function is $U(s)$ and the output is $Y(s)$. The natural frequency $w_0$ defines the rise-time $t_p$ and is set to equal to the swing time $t_s/2$. The input to the function is scaled step-function where the scaling is equal to the control-pitch value $\theta_c$. This filter is deployed using a discrete-time state-space function in C++. The state-space parameters are found using python's control toolbox.

## A.3 TRACKING RESULTS VISUALISED

In the Results section, we report the tracking results in Table 2. We plot the tracking results for the right front leg as the robot climbs the terrain which we can see in centre panel of Fig. 5. We plot the joint-angle output of the VAE and the actual commanded values from the WBC in Fig. 6. We see excellent tracking results even when the robot climbs the terrain initially where we see a negligible divergence between the two recordings.

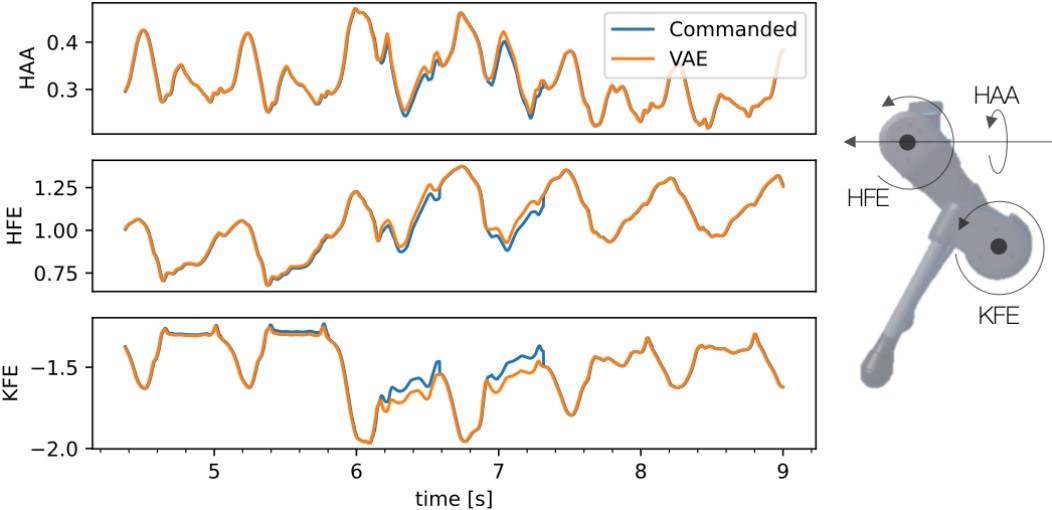

Figure 6: The joint-space trajectory in radians for the right front leg as it climbs terrain. The VAE output has key VAE and the WBC's commanded values use key commanded. We plot the leg next to the figure with annotations for the joint names.

