# OpenReview forum: "Gaitor: Learning a Unified Representation for Continuous Gait Transition and Terrain Traversal for Quadruped Robots"
_ICLR.cc/2024/Conference — Submitted to ICLR 2024_

### Official Review · Reviewer_keBn · 2023-10-26

**Soundness:** 4 excellent
**Presentation:** 3 good
**Contribution:** 3 good
**Rating:** 5
**Confidence:** 2

**Summary:**

Due to the requirements of the expected trajectory segmentations into a discrete set of skills, current methods mostly discretize commonalities among gait types, leading to less smooth transitions. To overcome these problems, the authors introduced Gaitor, leveraging a unified representation to capture the correlations among gait types. With the help of terrain encoding and a learned panner operation, Gaitor can take motion commands into account. Specifically, a modified variational autoencoder is applied to infer the robot's current and estimate the future state, which is also determined by the terrain latent space from the autoencoder. The evaluation is in simulated and real-world settings.

**Strengths:**

1. The writing, figures and tables are clear and understandable.
2. The proposed method is sound. Gaitor uses VAE and autoencoder to transfer the robot state and terrain information into latent space and does further operations like manual control or fuses information on the manifold, which is a good try to integrate multi-information.

**Weaknesses:**

1. The elements in the formula should be demonstrated more carefully, there are some units that first appear without description.
2. In this paper, there are few works to compare. Although it is firstly to do the task, there should be some way to compare the proposed methods with others, or it is hard to evaluate the performance.

**Questions:**

No more question.

---

> ### Author Response · Authors · 2023-11-15
> **Revisions after initial review**
>
> Dear Reviewer KEBN,
>
> Thank you for your review. We have found it most useful in improving our work.
>
> We have focused our attention on the presentation of the revised paper. In particular, we have restructured the methods section to improve the clarity of the approach and significantly revised Fig 1 and 2 so that the former acts as a deployment diagram, while the latter explains the training regime much more clearly. The revisions that we have made to our paper are in blue in the updated paper.
>
>
> We also contextualise the contributions of this work in relation to current art in particular Locomotion-Transformer. Gaitor is a representation learning method and does not use RL or exploration to solve locomotion problems. Instead, we learn a compact 2D representation for locomotion, which is trained using distinct gait. However, we discover that the approach exploits the phase correlations between the robot’s legs to create the representation meaning that, through traversal through the space, intermediate gaits are automatically discovered. To the best of our knowledge, this is the first method capable of generating novel intermediate gaits and does this without the use of RL or exploration of the state space. Furthermore, we show that these continuous transitions between gaits are dynamically feasible by deploying on the real Anymal C platform.
>
> We believe that our contributions have significant promise and ask for guidance as to how we can strengthen the revised manuscript so that it can showcase the contributions of our work.
>
> Kind regards,
>
> Anon.

---

> > ### Comment · Reviewer_keBn · 2023-11-19
> >
> > Hi,
> >
> > Thanks for your revision, and I am happy to see the presentation improvement. But the experiments are still my concern, since I can not see some comparison to show the effectiveness of Gaitor..
> >
> > Thanks,
> >
> > keBn

---

> > > ### Author Response · Authors · 2023-11-21
> > > **Response to Second Reviews**
> > >
> > > Dear Reviewer keBn,
> > >
> > > Thanks for taking the time to look over the revised manuscript.
> > >
> > > In the Results section, we draw comparisons between other methods in Table 1. We see that there are no other methods which are able to learn a compact representation for locomotion with which continuous gait transitions are automatically generated. Therefore, we are unable to directly compare our methods as they cannot produce this kind of behaviour.
> > >
> > > Our method differs considerably from prior work since our focus is to discover new intermediate gaits from distinct gait types such as trot, crawl and pace. Therefore, we present a representation learning methodology to infer gait transitions. We learn this representation using distinct gaits and then discover the gait transitions by exploring the learnt representation. This is vastly different from other works which learn policies for multiple gaits using reinforcement learning (RL) and in general leverage prior knowledge to learn a policy. For example, Yang et al., 2021 uses a predetermined contact schedule and cannot be used to discover new intermediate skills. Another method Locomotion-Transformer uses a large model with sufficient capacity to a multi-modal policy for locomotion. This method is an RL method and must explore the locomotion workspace to train the policy. Once trained, no new skills can be discovered since this method does not learn an explicit representation for locomotion.
> > >
> > > As such, we show the effectiveness of Gaitor by exploring the learnt latent-representation to explain the gait switching mechanism in latent space; we evaluate the performance of the perceptive system whilst climbing steps in the real-world; we show that the planner exploits the structure of the latent space to adjust the swing characteristics to help the robot climb steps; and finally we show that all the trajectories even the discovered gait transitions are dynamically feasible. We would be happy to consider augmenting this evaluation if there are specific aspects you could point us to (together with suitable baselines) which you feel might add value.
> > >
> > > Kind regards,
> > >
> > > Anon.

---

### Official Review · Reviewer_A1WD · 2023-10-27

**Soundness:** 2 fair
**Presentation:** 2 fair
**Contribution:** 2 fair
**Rating:** 5
**Confidence:** 3

**Summary:**

This paper presents a system that utilizes a VAE to encode different quadrupedal gaits into a continuous latent space. A planner then can utilize this latent space to continuously switch between gaits to accomplish different locomotion tasks.

**Strengths:**

1. The system is able to generate interpretable latent space that can be used to continuously switch between different gaits. It is also demonstrated on a real Anymal robot.

2. The analysis of the latent space demonstrates its interpretability.

**Weaknesses:**

I don't understand the approach section. It would be great if the authors could make the description clearer. See questions below.

**Questions:**

1. " All pose orientations are represented in tangent space." Can you specify what you mean here?

2. I don't quite understand why it is called a VAE instead of just a neural network that maps input to output. My understanding is that a VAE will try to reconstruct the input at its output, but that doe not seem to be the case here. Perhaps this is why I am very confused when reading the paper. My understanding is the proposed "VAE" learns to project the input to a latent space, which is then used to reconstruct the output (which is different from the input). VAE is also used when given the same input to generate multiple plausible outputs, in the context of robots, say given the same user command velocity, generates different gaits that can follow this velocity. I am not sure where is the variational part in the proposed vae.

3.  "The gait input is a label for each gait. This can be a one-hot encoding for each gait or a slider, where the three gaits, trot, crawl, and pace are [1, 0, −1]. The latter is recommended for its simplicity, but the one hot encoding produces the same results and
permits smooth transitions between gaits." I am confused by this sentence, who recommended the latter (and what is the latter?) And are you using this recommended option or are you using the one-hot encoding? And this performance predictor is never used again (I search for PP and performance predictor), what is it used for in the whole system?

4. I am not sure how the different components are used after training. My naive guess is the past trajectory is fed into the vae, and then output the future trajectory for robot control. The encoding of the inputs can be modified by the planner to adjust gaits. It would be nice to make it clearer how the system works.

5. Why is an additional planner needed to deal with terrains since terrain is already an input to the decoder and the training data for the decoder and planner are the same (or could be the same)? Maybe an ablation study is needed?

---

> ### Author Response · Authors · 2023-11-15
> **Revisions after initial review**
>
> Dear Reviewer A1WD,
>
> We have focused our attention on the presentation of the revised paper. In particular, we have restructured the methods section to improve the clarity of the approach and significantly revised Fig 1 and 2 so that the former acts as a deployment diagram, while the latter explains the training regime much more clearly. The revisions that we have made to our paper are in blue in the updated paper.
>
> We have significantly improved the clarity of the paper and ask you to revisit our contributions. Our work, Gaitor, is a representation learning method and does not use RL or exploration to solve locomotion problems. Instead, we learn a compact 2D representation for locomotion, which is trained using distinct gaits. However, we discover that the approach exploits the phase correlations between the robot’s legs to create the representation meaning that, through traversal through the space, intermediate gaits are automatically discovered. To the best of our knowledge, this is the first method capable of generating novel intermediate gaits and does this without the use of RL or exploration of the state space. Furthermore, we show that these continuous transitions between gaits are dynamically feasible by deploying on the real Anymal C platform. We ask for guidance as to how we can strengthen the revised manuscript so that it can showcase the contributions of our work.
>
>
>
>
> We address your individual remarks as follows:
>
> 1. We explain in the revised manuscript that the tangent space is a representation for the Lie-algebra of 3D rotations and is commonly used for robot control and state estimation. We include two citations as examples of how it is used in the wider literature also. Expressing coordinate transforms in the tangent-space has the benefit that non-linear composition of coordinate frames becomes linear vector algebra in the tangent space.
>
> 2. There are multiple reasons why we use a Variational Auto-Encoder (VAE) instead of an Auto-Encoder (AE). The first is that during training, the generative nature of the training process compresses our latent space meaning that we find a compact 2D representation for locomotion. This is crucial as properties of locomotion such as swing height and length are disentangled within the 2D space - while these are not explicitly varied parameters in the training set. This means we can solve for any locomotion problem using a trajectory in the 2D manifold. Complete details of this discussion can be found in prior work VAE-Loco. Thus, we refer the reader to this paper in Section 3.2, which discusses these findings in our paper.
> We use our learnt 2D latent-space to plan locomotion trajectories within. As we mention in the revised methods section 3, the latent space forms a solution space in which we select a single solution by traversing the space. Indeed, we are using the VAE to learn this manifold and as mentioned the generative training paradigm ensures that this space contains only the principal components necessary to reconstruct the locomotion trajectories.
>
> 3. We have simplified the gait label and removed reference to a one-hot encoding. This more clearly explains how we train the model and then during operation choose the desired gait.
> We also explain in greater detail what the Performance Predictors (PP) do. During training they help to structure the latent space: this was discovered in work done by prior authors in VAE-Loco. Secondly, we need to estimate the contact state of the robot and predict the future states so that the Whole-Body Controller (WBC) can take care of the contact-state dynamics. The PP predicts the future contact states and these are provided to the WBC.
>
> 4. We have clarified the deployment of Gaitor in two ways. Firstly, we have revised section 3.4 to explain this much more clearly. Secondly, we have revised Fig. 1 so that it explains the deployment process more clearly. Fig 1 is split into three panels which explain inference, planning and construction of the trajectories.
>
>
> 5. We clarify the contribution of the learnt planner in the revised Results Section 4. The planner exploits the latent-space structure  and is used to adjust the swing characteristics in response to terrain. Additionally, the modulation of the trajectory in latent-space is evident when we encode the expert trajectories.
>
>
> Kind regards,
>
> Anon.

---

> > ### Comment · Reviewer_A1WD · 2023-11-20
> > **response to rebuttal**
> >
> > Re: There are multiple reasons why we use a Variational Auto-Encoder
> >
> > The response does not address my concern that (in my opinion) the proposed method/network is not a VAE. Again, VAE is supposed to learn to reconstruct input, but in the proposed network, input and output are different.
> >
> > The other clarifications make the paper clearer.

---

> > > ### Author Response · Authors · 2023-11-21
> > > **Response to Second Reviews**
> > >
> > > Dear Reviewer A1WD,
> > >
> > > Thanks for taking the time to look over the revised manuscript.
> > >
> > > We are delighted that our modifications seem to have addressed all of your concerns apart from this minor disagreement.
> > >
> > > To clarify, the VAE reconstructs the current time step which is both in the input and the output as well as future states from the expert training trajectory during training. We use the term VAE in the same spirit as seminal works such as Autoencoding Variational Bayes (Kingma et al., 2022) in that we train our model by optimising the evidence lower-bound (ELBO), see Eq. 1 in our submission. We look forward to further discussing this point if required.
> > >
> > > Kind regards,
> > >
> > > Anon.

---

### Official Review · Reviewer_QD8G · 2023-10-30

**Soundness:** 3 good
**Presentation:** 1 poor
**Contribution:** 3 good
**Rating:** 3
**Confidence:** 5

**Summary:**

The paper proposes Gaitor, an imitation learning-based method for learning a continuous latent space of gaits for locomotion. A VAE latent space is trained on examples of expert trajectories. The paper uses the observation made by Mitchell'23 that good controllers can be created as elliptical trajectories in that latent space. The controller learns to predict an appropriate elliptical trajectory conditioned on terrain features. It is shown that the method can be used to automatically transition between gaits by representing the desired gait on a scale (trot = 1, crawl = 0, pace = -1) and continuously interpolating this variable. This experiment was performed in simulation, but transitioning between trot and crawl is possible on the real robot too.

**Strengths:**

- The paper proposes a promising method for gait transitions for locomotion
- The method doesn't use gait phases, instead implicitly discovering them from imitation data, which is a scalable approach.
- The empirical finding that it is possible to transition from trot to crawl is interesting.

**Weaknesses:**

- It is claimed that prior methods do not learn a shared representation between skills. However, Caluwaerths'23 proposes Locomotion-Transformer, which in fact does this, and reports similar results to this paper. Comparing to Locomotion-Transformer would help put the results in context.
- The presentation of the method is poor.
  - Fig 1 - none of the symbols are defined.
  - "Robot-specific encoding" section: $q_k, ee_k, \tau_k, \lambda_k, \dot{c}_k, \Delta c_k$ are all undefined.
  - "The latter is recommended for its simplicity, but the one hot encoding produces the same results". So which one was used in the experiments? Were all experiments performed with both?
  - "Terrain encoding" section: the terrain encoding is not defined in this section. Is terrain encoding same as $z_G$? Also $z_G$ is undefined.
  - Page 4 describes a method diagram in text which would be much more easily explained in a figure. Figure 1 presumably depicts the same information but is not helpful for understanding the method since Sec 3.1 doesn't reference the figure. The reader needs to guess what is the correspondence between the text and the figure. See e.g. Hafner'20 for an example of good presentation.
  - Eq 1. Y, U, s are undefined. The LTI function is never mentioned in the rest of the paper. Since all of the symbols are undefined, it is unclear whether this is used in the method at all, and if yes where.
  - "Training the VAE and Terrain AE" section: terrain encoder is undefined. Unclear how $\hat\theta_c(k)$ is produced. The VAE loss as far as I can tell doesn't depend on the terrain encoder, so it's unclear how the terrain encoder can be trained with those gradients. The terrain decoder is undefined.
  - There is a missing reference in the second line of Sec 2.
  - The citation for GECO is wrong. It is Rezende'18

Hafner'20, DREAM TO CONTROL: LEARNING BEHAVIORS BY LATENT IMAGINATION.
Rezende'18: Danilo Jimenez Rezende and Fabio Viola. Taming VAE, 2018.

**Questions:**

1. Why is it important that the method doesn't use gait phases as an inductive bias? A comparison to Yang'21 that showcases this difference between methods would strengthen the paper.
2. A discussion of a comparison to Locomotion-Transformer is crucial.
3. Overall, the paper is promising but I am unable to recommend accept due to poor presentation.

---

> ### Author Response · Authors · 2023-11-15
> **Revisions after initial review**
>
> Dear Reviewer QD8G,
>
> Thank you for your review, we have found the comments very useful.
>
> We have focused our attention on the presentation of the revised paper. In particular, we have restructured the methods section to improve the clarity of the approach and significantly revised Fig 1 and 2 so that the former acts as a deployment diagram, while the latter explains the training regime much more clearly. The revisions that we have made to our paper are in blue in the updated paper.
>
> We also contextualise the contributions of this work in relation to current art in particular Locomotion-Transformer. Gaitor is a representation learning method and does not use RL or exploration to solve locomotion problems. Instead, we learn a compact 2D representation for locomotion, which is trained using distinct gaits. However, we discover that the approach exploits the phase correlations between the robot’s legs to create the representation meaning that, through traversal through the space, intermediate gaits are automatically discovered. To the best of our knowledge, this is the first method capable of generating novel intermediate gaits and does this without the use of RL or exploration of the state space. Furthermore, we show that these continuous transitions between gaits are dynamically feasible by deploying on the real Anymal C platform.
>
> We also motivate why inferring the gait phase is of great importance. The discovery that the VAE can infer the gait phase is a fundamental discovery of our work, and in the revised results section, we explain that this representation directly results in the automatic discovery of novel gaits for gait transition. Furthermore, these new gaits are dynamically feasible.
>
>
> We also believe that our contributions have significant promise and have greatly improved the clarity of the work's presentation. Thus, we ask for suggestions as to how we can strengthen the revised manuscript so that it can showcase the contributions and impact of our work.
>
>
> Kind regards,
>
> Anon.

---

> > ### Comment · Reviewer_QD8G · 2023-11-20
> > **Changes too extensive**
> >
> > I appreciate the new version of the paper, which is much improved. However, a larger fraction of the content, perhaps around half has been updated in the revision. Such extensive changes are not suitable for a conference author response period since the paper now essentially needs to be evaluated from scratch. Because of this, I maintain my score and recommend resubmission to another conference.
> >
> > Other comments
> > - It is still unclear to me how the terrain encoder can be trained with robot decoder loss.
> > - I appreciate the discussion of locomotion-transfomer, but the paper still lacks quantitative experiments. The paper lacks any ablation analysis. In addition, if no suitable baselines exist, comparing to new naive baselines should still be possible. Alternatively, more extensive real world experiments could be presented with clear applications of the proposed method.

---

> > > ### Author Response · Authors · 2023-11-21
> > > **Response to Second Reviews**
> > >
> > > Dear Reviewer QD8G,
> > >
> > > As suggested by a number of the original reviews, we have endeavoured to improve the presentation by restructuring the methods section. The contributions, scientific content, overall structure, experiments in both simulation and hardware, introduction and abstract are unchanged. We therefore argue that the nature of the submission has not changed and the reviewer’s new stance, that our submission should no longer be considered for publication is not, in fact, justified. We note that none of the other reviewers have raised this as a concern and propose that this sort of revision is exactly the point of this phase of the review process.
> > >
> > > We request that the Area Chair mediate this concern.
> > >
> > > Regarding the stipulated “naive” baseline, we would appreciate insight into what specifically should be considered and how this would serve to further highlight the benefits of our method beyond what we have already included. Locomotion (especially in the real world) does not lend itself naturally to what one might consider naive approaches. As shown in Table 1 in the Results section, there are no comparable methods available that produce a representation which can discover continuous gait transitions other than our own.
> > >
> > > As mentioned in the paper, few works are able to generalise over multiple gait types. The methods which do tend to use reinforcement learning to learn a policy using explicit inductive biases. For example, Yang et al., 2021 uses a predetermined contact schedule to train a gait switching policy and operates in task-space. Our work is able to discover a joint-space trajectory and contact schedule for gait transition purely from the representation learnt using examples of the distinct gaits. Locomotion-transformer uses a model of suitable capacity to learn a multi-modal distribution conditioned on a terrain token to produce a policy suitable for traversing different types of terrain. This is not a representation learning model and uses RL during training to learn policies. In contrast, once our locomotion representation is learnt, we can generate new intermediate gaits just by traversing the structure of this space.
> > >
> > > Kind regards,
> > >
> > > Anon.

---

### Official Review · Reviewer_st69 · 2023-10-31

**Soundness:** 2 fair
**Presentation:** 2 fair
**Contribution:** 2 fair
**Rating:** 5
**Confidence:** 4

**Summary:**

The paper introduces Gaitor, a framework for learning effective representation of different gaits for quadrupedal locomotion and adaptive terrain traversal schemes. The model is composes of 3 segments: (1) a VAE which encodes history of robot states (proprioceptive) into latent embeddings and decodes to future sequence of states and footholds, (2) an AE which encodes local terrain heightmap filtered using LTI in frequency domain based on future footholds and decodes the same and (3) a planner MLP which takes the phase of the motion and terrain latent to output the radius of an elliptical trajectory in polar coordinates. The latent embedding of past trajectory is filtered to remove high-frequency components as most information is captured by the low-frequency components and then converted into polar form with phase angles. In the polar form, the radius predicted by the planner is used to modulate the latent space to adjust foot-step height and lengths. The proposed setup is demonstrated on a real-world scene.

**Strengths:**

Gaitor introduces an adaptive planner in the latent space of the VAE. It builds upon the advances introduced by VAE-loco on discretizing the latent space to represent different phases of the footsteps in a gait. Finally, the terrain embeddings are used to modulate the latent trajectories and hence the required footstep height and length for a given heightmap. This helps in building a framework for continuous transition between different quadrupedal gaits.

**Weaknesses:**

The paper is not written clearly. There is a lot of missing information and abuse of notations. Further, the authors do not justify their choices via suitable ablations which weakens the overall contributions of their method. Please answer the questions below:

How are the latent visualizations constructed specifically? From the material in the paper, it seems like only two lowest-variance (low-frequency, low noise) components of VAE latent embedding are selected and transformed into polar coordinates about the mean as center?

If only the selected components are modulated by the planner predictions of radius, are other components of the latent space discarded before input to decoder? If not, what happens if they are discarded? Else vice-versa? From the context, it seems those are high-frequency components and do not contain much information.

The inference pipeline is not clearly mentioned anywhere. The inputs are the history of states to the VAE which gives a latent trajectory representation. Now, how are the heightmap features calculated? The paper mentions “The height of the terrain at the future footholds are measured from this height map”. How do you get the future footholds? How are the predicted states used?

There has to be an appendix section clearly mentioning the construction of LTI transfer function for getting $y_k$ from $\theta_k$. What is $y_k$? How is $\theta_k$ constructed? How do you define angle between FL/FR and RR/RL? What is $w$ in equation (1)?
Without any definition, the authors have introduced the subscript “$c$” in equation (4)? Which when followed in section 3.2 becomes more confusing. What are $C$-discrete bins? What is $r^*$?

The paper is not very understandable at this point. However, it gives the readers an intuition of what is happening and how it is useful in understanding continuous transition between multiple gaits.

The paper contains remarks like “For example, it is possible to transition from trot to crawl to pace and the reverse only” without any reason.

**Questions:**

See weakness above.

---

> ### Author Response · Authors · 2023-11-15
> **Revisions after initial review**
>
> Dear Reviewer ST69,
>
> Thank you for your review, we have found the comments very useful.
>
> The consensus between reviewers is that though the soundness of the research is good, the presentation is poor. Therefore, we have focused on this area to improve the paper. In particular, we have restructured the methods section to improve the clarity of the approach and revised Fig 1 and 2 so that the former acts as a deployment diagram, while the latter explains the training regime much more clearly. The revisions that we have made to our paper are in blue in the updated paper.
>
> As a result of our modifications, we ask the reviewers to reevaluate our work and our contributions. Gaitor is a representation learning method and does not use RL or exploration to solve locomotion problems. Instead, we learn a compact 2D representation for locomotion, which is trained using distinct gait. However, we discover that the approach exploits the phase correlations between the robot’s legs to create the representation meaning that, through traversal through the space, intermediate gaits are automatically discovered. To the best of our knowledge, this is the first method capable of generating novel intermediate gaits and does this without the use of RL or exploration of the state space. Furthermore, we show that these continuous transitions between gaits are dynamically feasible by deploying on the real Anymal C platform.
>
> Before we address your specific comments, we ask for guidance as to how we can strengthen the revised manuscript so that it can showcase the contributions of our work.
>
>
>
> Responses to your specific queries:
>
> To address your first comment, we explain more clearly how the latent-space images are created in Section 3.2 and refer the reader to a long-form recipe for creating these that we used from prior work found in VAE-Loco. We explain that a 2D slice in latent space is sufficient to reconstruct locomotion trajectories to a high fidelity.
>
>
> We explain the inference pipeline in much greater detail in the revised Section 3.4 and have significantly revised Fig. 1 to explain how the inference works whilst Gaitor is deployed on the real robot. We also have a new Fig. 3 which explains how the terrain is processed before it is encoded into latent space. This figure is referred to in Section 3.1, where we also explain how we predict the future footholds.
>
> We have addressed weaknesses 4 and 5 together. We have clarified how the VAE learns to represent the locomotion data in Section 4 (Gait Switching subsection). This more clearly explains how the VAE infers the relative phases between all the legs to learn a compact representation for the multiple gaits. This structure in the latent space explains why gait transitions can only move from trot to crawl and then to pace.
>
> Thank you for your reviews. I hope to hear your thoughts on the updated manuscript soon.
>
> Kind regards,
>
> Anon.

---

### Author Response · Authors · 2023-11-15
**Summary of initial reviews**

We thank all the reviewers for their useful comments.

The consensus between reviewers is that though the soundness of the research is good, the presentation is poor. Therefore, we have focused on this area to improve the paper. In particular, we have restructured the methods section to improve the clarity of the approach. The revised method section is split up into three stages for training all the components and a deployment stage. The three training stages require firstly learning a latent-space for locomotion in which planning can take place. Secondly, the second stage requires inspecting the learnt representation to discover useful structure. This structure informs design decisions for the third planning in latent-space stage. To highlight the three stages, we have revised Fig. 2 to depict the three training phases and adjusted Fig. 1 so that it can be used to both introduce the work and also to show the deployment process.

We also contextualise the contributions of this work in relation to current art in particular Locomotion-Transformer. Gaitor is a representation learning method and does not use RL or exploration to solve locomotion problems. Instead, we learn a compact 2D representation for locomotion, which is trained using distinct gaits. However, we discover that the approach exploits the phase correlations between the robot’s legs to create the representation meaning that, through traversal through the space, intermediate gaits are automatically discovered. To the best of our knowledge, this is the first method capable of generating novel intermediate gaits and does this without the use of RL or exploration of the state space. Furthermore, we show that these continuous transitions between gaits are dynamically feasible by deploying on the real Anymal C platform.

---

### Meta-Review · Area_Chair_iyU9 · 2023-12-06

**Metareview:**

*Summary*: This paper presents Gaitor, a framework that uses VAE to encode various quadrupedal gaits into a continuous latent space. Then, a planner utilizes this learned latent space to continuously switch between gaits to achieve versatile locomotion skills. Simulation and real experiments on ANYmal C platform show the effectiveness of the proposed approach.

*Strength*: (1) The idea of learning a shared representation of all gaits is very interesting and novel. (2) The proposed latent space learning method doesn't need RL-style trial and error.

*Weakness*: (1) Need more justifications for why continuously switching between various gaits is important. In particular, it seems that other RL-based or MPC-based baselines will also perform well in this paper's experiments. In other words, this paper will be significantly stronger if it finds some applications that require smooth and continuous gait transitions. (2) Writing and presentation need improvement.

**Justification For Why Not Higher Score:**

See the weakness part.

**Justification For Why Not Lower Score:**

See the strength part.

---

### Decision · Program_Chairs · 2024-01-16

Reject